# Transfer Rule Learning over Large Knowledge Graphs

## ABSTRACT

Logical rules have been widely used for expressing schema knowledge in various practical applications. It is infeasible to handcraft rules from large knowledge graphs (KGs) and thus many methods have been proposed for learning rules automatically from KGs. However, it is largely ignored how to extract rules in a (target) KG from rules that already exist in some other (source) KGs. In this paper, we propose a framework for KG rule learning based on transfer learning. A major challenge for establishing such a framework is that a suitable alignment mechanism is required for mapping certain subgraph structures between predicates in the source KG and the target KG. Hence, our framework provides a new method for predicate mapping based on graph-structural similarity. The proposed framework can be used as a standalone rule learner but more importantly, it paves a new way for enhancing the state-of-the-art rule learners for large KGs. Extensive experiments are conducted to evaluate the new approach to rule learning, which shows that rules in smaller KGs can be effectively transferred to a large KG.

## CCS CONCEPTS

• **Information systems** → **Semantic web description languages**.

## KEYWORDS

knowledge graph, transfer learning, rule learning.

**ACM Reference Format:**
. 2024. Transfer Rule Learning over Large Knowledge Graphs. In *TheWebConf '25: The Web Conference, April 28–May 02, 2025, Sydney, Australia.* ACM, New York, NY, USA, 9 pages. https://doi.org/10.1145/nnnnnnn.nnnnnnn

## 1 INTRODUCTION

Knowledge graphs (KGs) offer a flexible and powerful way to represent background knowledge, by organising objects of interest as entities and interconnecting them through various relations (a.k.a. predicates) [14]. A KG can be represented as a set of triples of the form (*subject*, *predicate*, *object*), where *subject* and *object* are entities associated by the *predicate*, such as (*M. Jordan*, *bornIn*, *New York*) saying *M. Jordan* was born in *New York*. Numerous KGs have been constructed in academia and industry, such as YAGO [28], DBpedia [1], Wikidata [32], and NELL [5], which have been adopted as essential components for intelligent Web search, recommender systems, and question answering.

Due to the large scales of KGs and diversity of knowledge contained in them, reasoning over KGs is a challenging task. Rules can represent higher-order patterns of KGs, and have been widely used

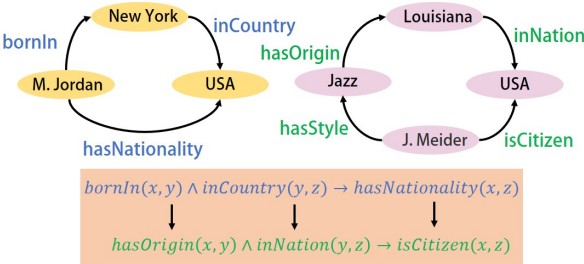

**Figure 1: Structural patterns and predicate mappings for rule transfer.**

for expressing schema knowledge in various practical reasoning systems. Especially, rules can be used to derive new information in a KG, which is useful for building and expanding KGs, including providing solutions to explainable link prediction [8, 10, 19, 22]. For example, a rule of the form

$$0.8 : bornIn(x, y) \land inCountry(y, z) \rightarrow hasNationality(x, z) \quad (1)$$

says a person $x$ born in a city $y$ of a country $z$ is likely to be a national of $z$ (with a confidence degree of 0.8). If we know that M. Jordan was born in New York, which is a city in the United States, then we can derive that M. Jordan is a national of the United States (with a relatively high confidence).

It is infeasible to handcraft rules from large KGs and thus many methods have been proposed for automatically learning rules from KGs [7, 8, 10, 19, 22, 23, 25, 33]. A basic idea in these methods is to exhaustively search useful structural patterns over the entire KG, which is computationally expensive. We call such methods standard rule learners, i.e., a rule learner that does not use transfer learning. However, it is largely ignored how to extract rules in a (target) KG from rules that already exist in some other (source) KGs. That is, to transfer existing rules from a source KG $\mathcal{K}'$ to a (usually large) target KG $\mathcal{K}$ by setting up an alignment between the predicates of $\mathcal{K}'$ and $\mathcal{K}$. Yet this idea may not work if the alignment between $\mathcal{K}'$ and $\mathcal{K}$ fails to capture certain subgraph structures in the KGs.

Consider the source and target KGs (respectively, on the left and right) in Figure 1. Based on the meanings of these entities and predicates, we could have an alignment of the predicates in the KGs as indicated by the arrows between the two rules. As a result, the rule (1) could be transferred to a rule

$$hasOrigin(x, y) \land inNation(y, z) \rightarrow isCitizen(x, z),$$

which is clearly unintuitive. For example, Jazz originated in Louisiana and Louisiana is in the USA, then the rule would derive that Jazz is a citizen of the USA. This example shows that a semantic-based predicate alignment does not always work in rule transfer and thus a suitable framework for transfer rule learning is required.

Yet, to our best knowledge, transfer rule learning over KGs has not received much attention, with few exceptions [21, 29]. These approaches use existing KG-embedding models to capture subgraph structures in KGs to transfer rules of source KGs to the target KG.

However, such KG-embedding models often focus on entities and can easily be impacted by the (often significantly) different entity distributions between the source and target KGs. On the other hand, to transfer rules, capturing the structural patterns of how predicates correlate to each other (i.e., connected via entities) is more crucial. For example, in Figure 1, predicates *bornIn* and *hasNationality* in the source KG are correlated, whereas *hasOrigin* and *isCitizen* in the target KG are not.

In this paper, we first propose a novel predicate mapping method to capture certain subgraph structural information that is essential for rule transfer. Our method is based on a new predicate-centric graph representation and a graph embedding model GDV [12]. While KG embeddings have been extensively studied to capture structural features of KGs, we note that existing KG embeddings are centred around individual entities and fail to naturally meet the requirement of characterising predicate correlations. To remedy this shortcoming, for each predicate $p$, we construct a new graph called predicate-correlation graph $G_p$ from the given KG. The basic idea is to represent predicates related to $p$ as vertices and their correlations as edges in $G_p$. Such graphs capture critical structural features for rule transfer and are invariant to entity distributions. Then, the embedding of $p$ is defined as the GDV matrix of $G_p$. In this way, the required graph-structural similarity is captured.

We develop a transfer rule learner TRuLer (**T**ransfer **Ru**le **L**earn**er**) based on the new predicate mapping. In several different settings, we evaluate it with state-of-the-art rule learners including an existing transfer rule learners. Our experiments show that TRuLer is capable of learning a significant amount of rules in a reasonable timeframe, making it a strong competitor to standard rule learners such as AMIE3 [16]. The quality of transferred rules is also high and can be used to complement the state-of-the-art rule learners. We also show that TRuLer can effectively transfer rules with varying sources and across different domains.

## 2 RELATED WORK

In this section, we discuss existing works that are closely related to the paper.

### 2.1 Rule Learning

Existing rule learning approaches for large KGs mostly fall into three major groups, path-based [7, 19, 23], ILP-based [8, 10], and neural-based [9, 22, 24, 25, 33]. Path-based approaches learn rules that resemble paths in KGs. Paths in KGs are considered important features for KG predictions, for instance, the PRA algorithm [17] predicts whether two entities are connected by a predicate by exploring paths between them through a random walk. AnyBURL [19] explores paths in KGs to generate rules. As examining all paths in large KGs is infeasible, AnyBURL adopts an anytime algorithm (i.e., it needs to be configured when to stop) and often can only explore a (small) portion of potential paths in a reasonable time. Based on Inductive Logic Programming (ILP) research, AMIE and its extensions [10, 16] traverses the rule space through rule refinement and applies several heuristics to reduce the search space. While it can perform an exhaustive search over small or medium-sized KGs and produce high-quality rules, such a search is quite expensive for large-scale KGs. Hence, methods are proposed to generate

candidate rules through neural networks. In particular, RLvLR [22] generates candidate rules through latent representations (called embeddings) produced by neural networks and uses a KG sampling method to reduce the computation cost for embeddings. It is the first embedding-based rule learner that can scale over large KGs Wikidata and DBpedia. Some recent works also focus on rule-based link prediction using neural networks [9, 24, 25, 33]. For example, RNNLogic [24] treats logical rules as latent variables for training rule generators and inference predictors, and RLogic [9] recursively splits rule paths into atomic models and proposes a rule-scoring function based on predicate representation to filter rules. Many of these approaches cannot handle large KGs due to the expensive training of neural networks.

Transfer rule learning over KGs has been recently studied [21, 29], which utilises embeddings to map predicates between (multiple) source KGs and the target KG. Due to the high cost of computing the embeddings and mappings, they are only suitable for relatively small KGs and their accuracy is not high. In this paper, we aim to tackle these challenges by proposing a novel transfer rule learner that is scalable and has competitive accuracy to major KG rule learners.

### 2.2 Predicate Alignment

KG alignment aims to discover entities (or predicates) with different names in two different KGs that essentially refer to the same objects (resp., relationships). Most ontology alignment techniques rely on a large number of alignment seeds to guide the alignment process, known as supervised ontology alignment methods [2, 15, 30, 36]. Recently, unsupervised ontology alignment methods have become increasingly popular, such as [18, 20, 27, 31]. For example, SEU [18] uses word vectors and character vectors of entity or relationship names to align them, relying entirely on the lexical-semantic information of the ontology. FGWEA [31] combines ontology semantics and KG structural features, and aligns them using the fused Gromov Wasserstein (FGW) distance. These approaches focus on semantic similarities (i.e., similar meaning) of the predicates, but cannot fully capture graph-structural similarities which are critical for rule learning. Unlike these approaches, we use predicate embeddings to effectively capture and efficiently measure graph-structural similarities to map predicates between KGs.

## 3 PRELIMINARIES

In this section, we briefly introduce some basics of knowledge graphs and rule learning, as well as fixing some notations to be used later.

### 3.1 Knowledge Graphs and Rules

A *knowledge graph* (*KG*) is a directed multi-relational graph, often expressed as a set of *triples* of the form $(e, p, e')$, where $e, e'$ are *entities* and $p$ is a *predicate*. Let $\mathcal{E}$ and $\mathcal{P}$ be respectively the sets of entities and predicates in the KG $\mathcal{K}$. Following the tradition in knowledge bases, we denote a triple a triple $(e, p, e')$ as $p(e, e')$, and also refer to it as a *fact* in the KG. For a predicate $p$, $p^-$ denotes its *inverse*, i.e., triple $p^-(e', e)$ is equivalent to $p(e, e')$. Let $\mathcal{P}^* = \mathcal{P} \cup \{p^- \mid p \in \mathcal{P}\}$.

There has been much interest in learning first-order Horn rules from KGs [8, 10, 22, 23]. A first-order Horn rule is of the form

$$B_1 \wedge B_2 \wedge \cdots \wedge B_n \rightarrow H, \qquad (2)$$

where $H$ (resp., each $B_i$ for $1 \leq i \leq n$) is of the form $p(t, t')$ with $p \in \mathcal{P}$ (resp., $p'(t, t')$ with $p' \in \mathcal{P}^*$) and each of the $t, t'$ is a variable or an entity. $H$ is the *head* of the rule and the set $\{B_1, B_2, \cdots, B_n\}$ is the *body* of the rule. The *length* of the rule is $n$.

The plausibility of candidate rules is commonly assessed using the metrics *support*, *standard confidence*, and *head coverage* [8, 22], or their variants [10]. To assess the plausibility of candidate rules, for a rule $r$ of the form (2) with its head being $p(t, t')$, $ins_H(r)$ consists of all the pairs of entities $e, e' \in \mathcal{E}$ such that $p(e, e')$ occurs in the KG. Similarly, $ins_B(r)$ consists of all the pairs of entities $e, e' \in \mathcal{E}$ such that there is a way to substitute the variables in $r$ mapping $t, t'$ to $e, e'$ and that all the substituted body atoms of $r$ occur in the KG. Then, the *support* of $r$ is defined as $|ins_H(r) \cap ins_B(r)|$ [10]. That is, the support of $r$ is defined as the number of entity pairs that satisfy both the head and the body of $r$. The *standard confidence* (*SC*) and *head coverage* (*HC*) of $r$ are defined as follows

$$sc(r) = \frac{|ins_H(r) \cap ins_B(r)|}{|ins_B(r)|} \text{ and } hc(r) = \frac{|ins_H(r) \cap ins_B(r)|}{|ins_H(r)|}$$

Hence, SC is the normalisation of support through the number of entity pairs that satisfy the body, while HC is the normalisation of support through the number of entity pairs that satisfy the head. The higher the values are the more plausible the rule is.

*Link prediction* is the task that given an entity $e \in \mathcal{E}$ and a property $p \in \mathcal{P}^*$, to predict entities $e'$ such that $p(e, e')$ is plausible. Unlike embedding-based approaches that rank the possible entities $e'$ via scoring functions, a rule-based approach tries to derive plausible facts $p(e, e')$ by applying the learned rules to the existing facts in the KG. The ranking of the derived fact is obtained from the confidence degrees of the rules deriving it.

# 4 OUR APPROACH

In this section, we present a new transfer rule learning model TRuLer (**T**ransfer **Ru**le **L**earn**er**). The basic idea is to adapt (or transfer) the rules from the source domain to form rules of the target domain. Our model consists of four major components: (1) predicate-correlation graph (PCG) construction, where we construct graphs centred around predicates based on their structural features, (2) predicate embedding, where embeddings of predicates are generated from PCGs, (3) predicate mapping, where we align predicates in the source and target KGs according to predicate embeddings, and (4) rule transfer and validation, where we transfer rules from the source KG to the target KG and validate the transferred rules.

Figure 2 shows an overview of our approach. Consider a source KG $\mathcal{S}$ with a collection of source rules $\mathcal{R}_{\mathcal{S}}$, and a target KG $\mathcal{T}$ (with an empty set of rules). $\mathcal{R}_{\mathcal{S}}$ can be crafted by the user for the source KG $\mathcal{S}$ or learned from an off-the-shelf rule learner, such as AnyBURL [19]. To transfer the source rules to the target, we construct a mapping $\mathcal{M}$ from the set of predicates in $\mathcal{S}$, denoted $\mathcal{P}$, to the set of predicates in $\mathcal{T}$, denoted $\mathcal{Q}$. This is achieved by constructing a predicate-correlation graph (PCG) for each predicate and using graph embeddings to align predicates between $\mathcal{P}$ and $\mathcal{Q}$.

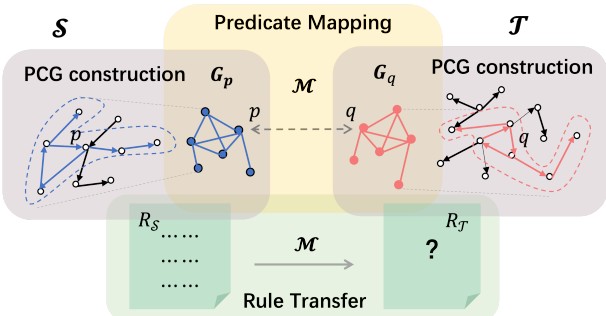

**Figure 2: An overview of our approach.**

However, existing models of embedding for KGs do not naturally meet this requirement and thus, we introduce a new embedding based on graph degree vectors (GDV) [12]. Then we will be able to transfer rules from the source domain to the target domain.

The technical details in each of the modules will be explained in the rest of this section.

## 4.1 Local Predicate-Correlation Graphs

As discussed previously, in the setting of transfer rule learning we need a predicate mapping that preserves certain graph patterns between the source and target KGs. Such a structural pattern aims to provide a template for candidate rules. Most existing methods of ontology/KG alignment and mapping are designed to establish relationships between lexically or semantically similar predicates [20, 27] but are not suitable for mapping predicates with similar local graph structures. For example, they can map the predicate *country* in a KG to the predicate *nation* in another KG as they are synonyms, yet such a mapping is not necessarily useful for rule transfer as the two predicates may occur in very different graph structures. On the other hand, as Figure 3 shows, the predicates *country* and *release_medium*, although semantically very different, may play similar structural roles in their respective KGs and thus occur in rules of similar shapes.

Consider a rule

$$designer(x, x_1) \wedge manufacturer(x_2, x_1)$$
$$\wedge\ country(x_2, y) \rightarrow country(x, y). \quad (3)$$

which says if $x$ is the designer of product $x_1$, $x_2$ is the manufacturer of $x_1$, and $x_2$ is in country $y$, then $x$ is in the same country $y$. The graph patterns captured by the rule include, for instance, predicates *designer* and *manufacturer* being correlated via instances of $x_1$ and *designer* and *country* being correlated via instances of $x$.

To capture such correlations between predicates, inspired by [6], we define two predicates $p_1$ and $p_2$ in KG $\mathcal{K}$ to be *correlated* if they satisfy one of the six conditions for some entities $e_1, e_2$ and $e_3$ in $\mathcal{K}$:

- tail-head (TH): $p_1(e_1, e_2)$ and $p_2(e_2, e_3)$ are in $\mathcal{K}$ with $e_1 \neq e_3$;
- head-tail (HT): $p_1(e_1, e_2)$ and $p_2(e_3, e_1)$ are in $\mathcal{K}$ with $e_1 \neq e_3$;
- head-head (HH): $p_1(e_1, e_2)$ and $p_2(e_1, e_3)$ are in $\mathcal{K}$ with $e_2 \neq e_3$;
- tail-tail (TT): $p_1(e_1, e_2)$ and $p_2(e_3, e_2)$ are in $\mathcal{K}$ with $e_1 \neq e_3$;
- loop (LP): $p_1(e_1, e_2)$ and $p_2(e_2, e_1)$ are in $\mathcal{K}$;

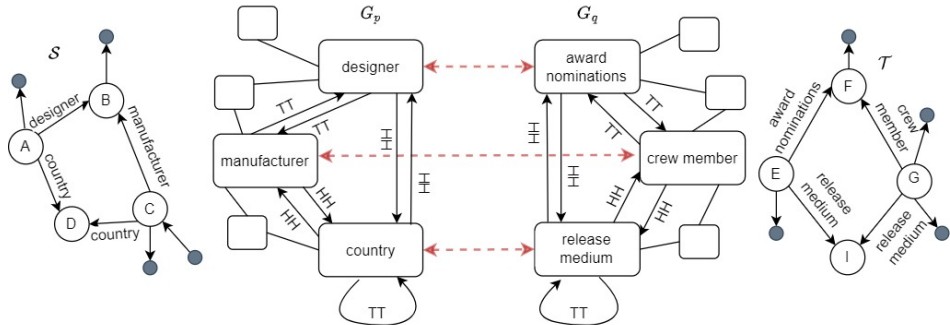

**Figure 3: An example of structural similarity captured by PCGs.**

- parallel (PL): $p_1(e_1, e_2)$ and $p_2(e_1, e_2)$ are in $\mathcal{K}$.

We say that $p_1$ is an $\epsilon$-neighbour of $p_2$ for $\epsilon \in \{TH, HT, HH, TT, LP, PL\}$, and the *support* of such a relationship is the number of different pairs of the corresponding facts defined as above. The *predicate-correlation graph (PCG)* of a KG $\mathcal{K}$ is defined as a directed graph whose vertices consist of all the predicates in the KG, and there is an edge from $p_1$ to $p_2$ with the label $\{\epsilon : n_\epsilon\}$ if $p_1$ is an $\epsilon$-neighbour of $p_2$ with a support $n_\epsilon$. Note that support values are not considered in [6] but are useful in our case.

A global PCG for the whole KG is usually too large to be efficiently constructed. Hence, we introduce the notion of local PCGs. That is, for a predicate $p$, we extract a module of the KG that captures the correlation information on $p$, using the sampling method in [22], and then build the local PCG $G_p$ as a module of the given KG, called the *PCG for $p$*. For example, Figure 3 shows the modules for predicates *country* and *release_medium* together with the corresponding PCGs.

Besides obtaining PCGs from (modules of) KGs, we can also construct PCGs directly from a collection of rules. The advantage of this rule-based construction for local PCGs is that rules from the source can be transferred to the target without accessing specific triples. This is useful in some scenarios. For instance, due to confidentiality concerns, certain facts in a KG may not be accessible. For a predicate $p$, suppose $\mathcal{R}_p$ consists of rules with $p$ occurring in their heads. Note that in each rule, the predicates are correlated to each other in a similar way as in KGs, being connected via variables instead of entities. Consider rule (3), the support of the TT correlation between *designer* and *manufacturer* can be approximated as $|ins_B(r)|$, whereas that for the HH correlation between *designer* and *country* can be $|ins_H(r) \cap ins_B(r)|$.

The PCG of predicate $p$ obtained from $\mathcal{R}_p$, also denoted $G_p$, can be defined in a similar way as for a KG, by replacing the entities $e_1, e_2, e_3$ in the definition with variables $x_1, x_2, x_3$. For each $\epsilon$-neighbour $p_1$ of $p_2$, the support is $\sum_{r \in \mathcal{R}_p} \sum_{1 \le i \le n_r^\epsilon} e_i$, where $n_r^\epsilon$ is the number of $\epsilon$-correlation between $p_1$ and $p_2$ in $r$ and $e_i$ is the support, defined as $e_i = |ins_B(r)|$ if both $p_1$ and $p_2$ occur in the body of $r$, and otherwise if one of them is in the head, $e_i = |ins_H(r) \cap ins_B(r)|$. The PCG $\mathcal{R}_p$ of $p$ can be seen as an approximation of that obtained from the KG. Such an approximation

is useful as it directly reflects the graph pattern information represented in the rules to be transferred and its computation is often much more efficient compared to that from the (modules of) KG.

## 4.2 Predicate Embedding

To develop a suitable mechanism for transferring rules from a source to the target, we need an embedding method that can characterise certain topological properties of predicate-correlation graphs. Such an embedding method needs to be different from existing KG embedding methods in the following aspects. First, existing KG embedding methods need to embed each entity in the KG, and the predicate embeddings are obtained from entity embeddings (via loss functions defined from KG triples). For rule transfer, we only need to embed predicates and their embeddings should capture how their graph-structural features, i.e., how they correlate with other predicates. Also, while KGs often have a huge number of entities, the numbers of predicates are often much smaller. Hence, predicate-correlation graphs are often significantly smaller than the original KGs. Moreover, the number of predicates occurring in a rule is bounded by the rule length. To transfer rules of a maximum length $l$, our embeddings need to capture topological similarities between sub-graphs (in predicate-correlation graphs) with maximum $l$ vertices. Finally, unlike other graph embeddings, our graph embedding pays less attention to the vertex degrees but focuses on the topological positions of vertices in the sub-graphs.

Based on these considerations, our embedding method is based on the framework of *graphlet degree vectors (GDVs)* [12, 34], where vertex features are modelled by all the *graphlets* where the vertex occurs. In a large graph $G$, for a subset $V$ of the vertices in $G$, a *graphlet* induced by $V$ is a subgraph of $G$ whose vertices are $V$ and whose edges are all those in $G$ with both endpoints being in $V$. For example, the subgraph with 3 vertices *designer*, *manufacturer*, and *country* in Figure 3 form a graphlet. As existing rule learners typically learn rules with lengths smaller than or equal to 5, GDV graphlets with at most 5 vertices are sufficient to capture the graph patterns we need. Within a graphlet, *orbits* are different topological positions that a vertex can occur in the graphlet. For example, for a graphlet with 2 vertices, there is only one orbit, indexed as orbit-0, as the topological positions of both vertices are identical; for a graphlet with 3 vertices forming a chain, there are two orbits, one on the endpoint, indexed orbit-1, and one in the middle, indexed

orbit-2; and for a triangle-shaped graph with 3 vertices, there is only one orbit indexed orbit-3.

To learn rules with lengths smaller than or equal to 5, we consider graphlets with up to 5 vertices. There are 29 topologically different graphlets with at most 5 vertices, and totally 73 possible orbits in these graphlets. As a result, the GDV of a vertex in a graph is represented as a vector of 73 dimensions, where the $i$-th dimension records the number of graphlets in which the vertex is found in orbit-$i$. For example, for a triangle-shaped graph with 3 vertices, each vertex appears in orbit-3 only once (i.e., one graphlet which is the whole graph), so the fourth component of its GDV vector is 1. Also, each vertex is connected to the other two vertices, so it appears in orbit-0 twice (i.e., in two graphlets), and the first component of its GDV vector is 2. So each vertex has a GDV of the form $[2, 0, 0, 1, 0, \ldots, 0]$.

For each PCG $G$, we can obtain a GDV embedding $\mathbf{G} \in \mathbb{R}^{n \times d}$ by stacking the vertex vectors, from the simplified graph of $G$ by omitting the labels of the edges in $G$, where $n$ is the number of vertices in $G$ and $d$ is the dimension of a GDV. The labels of the edges are stored in a tensor $\alpha$ and processed later. Let $\mathbf{G}[i, j]$ $(1 \leq i \leq n, 1 \leq j \leq d)$ denote the value indexed by $i$ and $j$ in $\mathbf{G}$. Due to the (potentially) significant difference between the sizes of the source and the target PCGs, the GDV embeddings generated from them need to be normalized as follows,

$$\hat{\mathbf{G}}[i, j] = \frac{\mathbf{G}[i, j] - \min(\mathbf{G}[\cdot, j])}{\max(\mathbf{G}[\cdot, j]) - \min(\mathbf{G}[\cdot, j])}. \qquad (4)$$

Let $\mathbf{g}_i$ be the $i$-th row of $\hat{\mathbf{G}}$, and it encodes the topological information of the $i$-th vertex $v_i$ in $G$, without labels of the edges. To capture the labels of the edges, $\alpha_{i,j,\epsilon}$ $(1 \leq i, j \leq n)$ denotes the support for the $\epsilon$-neighbour $v_j$ of vertex $v_i$ in $G$.

Inspired by R-GCN [26], we aggregate the encoding of neighbours to form the embedding of each vertex. Let $\mathbf{h}_i^0 = \mathbf{g}_i$, and

$$\mathbf{h}_i^k = \mathbf{h}_i^{k-1} + \sigma \Big( \sum_{\epsilon} \sum_{j \in N_{i,\epsilon}} \frac{\alpha_{i,j,\epsilon}}{\sum_{j \in \mathcal{N}_{i,\epsilon}} \alpha_{i,j,\epsilon}} \cdot \mathbf{h}_j^{k-1} \Big) \qquad (5)$$

where $\mathcal{N}_{i,\epsilon}$ consists of the $\epsilon$-neighbours of vertex $i$ in $G$ and $\sigma$ is an element-wise activation function. While various activation functions can be used here, we use ReLU in our model for its high computational efficiency.

We define $\mathbf{p}_i = \mathbf{h}_i^0 \oplus \mathbf{h}_i^1 \oplus \cdots \oplus \mathbf{h}_i^K$ be the embedding of predicate $p_i$, where $\oplus$ is vector concatenation and $K$ is the number of iterations. The similarity between two predicates $p$ and $q$ can be measured by

$$sim(p, q) = exp[-\|\mathbf{p} - \mathbf{q}\|_2^2]. \qquad (6)$$

## 4.3 Predicate Mapping and Rule Transfer

For predicate mapping, a naive method that calculates the similarity between each pair of predicates in respectively the source and target KGs is rather time-consuming and inefficient. For computation efficiency, we adapt the embedding-based graph alignment module of REGAL [11] to align the source and target predicates. Unlike REGAL, which uses simple vertex degrees for graph embeddings and cannot reflect the required structural similarity, we use our predicate embeddings instead. Following REGAL, our predicate alignment method uses a special data structure called k-d tree and an efficient nearest neighbour algorithm. We store the embedding

of each predicate in the source KG $\mathcal{S}$ in the k-d tree, and then traverse the predicates in the target $\mathcal{T}$ and evaluate the similarity between their embeddings in the k-d tree.

The predicate alignment computes a mapping $\mathcal{M}$ between $\mathcal{P}$ and $Q$, where each predicate $p \in \mathcal{P}$ is mapped to at least one predicate $q \in Q$ based on the similarity score $sim(p, q)$. If $p$ is mapped to $q$ in $\mathcal{M}$, it is denoted $(p, q) \in \mathcal{M}$. It is possible for source predicates to be mapped to a few "popular" target predicates, which would lead to a large number of repetitive rules after the transfer. Hence, we use a mechanism to ensure the target predicates are evenly distributed in the mapping $\mathcal{M}$.

Given a collection of source rules $\mathcal{R}_{\mathcal{S}}$ and a target KG $\mathcal{T}$, the rules in $\mathcal{R}_{\mathcal{S}}$ can be transferred to $\mathcal{T}$ via the mapping $\mathcal{M}$. For example, from each source rule of the form $p_1(x_1, y_1) \wedge p_2(x_2, y_2) \wedge \cdots \wedge p_n(x_n, y_n) \rightarrow p(x, y)$, we can obtain a candidate rule in the target domain $q_1(x_1, y_1) \wedge q_2(x_2, y_2) \wedge \cdots \wedge q_n(x_n, y_n) \rightarrow q(x, y)$ with $(p, q) \in \mathcal{M}$ and $(p_i, q_i) \in \mathcal{M}$ for $1 \leq i \leq n$. The candidate rules obtained from rule transfer can still contain noises and thus will be validated on the target KG through their SC and HC scores.

This method is specified in Algorithm 1.

---

**Algorithm 1** Transfer Rule Learning

---

**Input:** A set of source rules $\mathcal{R}_{\mathcal{S}}$ and a set of predicates $\mathcal{P}$, a target KG $\mathcal{T}$ with a set of predicates $Q$
**Output:** A set of rules $\mathcal{R}_{\mathcal{T}}$ on $Q$

1: **for all** $p \in \mathcal{P}$ **do**
2:     $G_p := \text{PCG}(\mathcal{R}_p)$
3:     $\mathbf{p} := \text{embedding}(G_p)$
4: **end for**
5: **for all** $q \in Q$ **do**
6:     $\mathcal{K}_q := \text{sampling}(\mathcal{T}, q)$;
7:     $G_q := \text{PCG}(\mathcal{K}_q)$
8:     $\mathbf{q} := \text{embedding}(G_q)$
9: **end for**
10: $\mathcal{M} := \text{mapping}(\{\mathbf{p}\}_{p \in \mathcal{P}}, \{\mathbf{q}\}_{q \in Q})$
11: $R_{\mathcal{T}} := \text{transfer}(\mathcal{R}_{\mathcal{S}}, \mathcal{M})$
12: $R_{\mathcal{T}} := \text{validate}(\mathcal{R}_{\mathcal{T}})$
13: **return** $\mathcal{R}_{\mathcal{T}}$

---

In Algorithm 1, lines 1 – 9, a rule-oriented embedding is generated for each source predicate and each target predicate, through sampling and PCG construction. Then, in line 10, the mapping is constructed. Finally, in lines 11 – 12, the source rules are transferred to the target via the mappings, and the candidate rules are validated by their SC and HC over the target KG.

## 5 EXPERIMENTS

We conducted extensive experiments to evaluate TRuLer under various settings and compare it with existing transfer rule learners and standard rule learners. In particular, we evaluate the scalability of TRuLer, i.e., whether our transfer rule learning can handle large-scale KGs as existing rule learners like AMIE3, as well as the quality of rules learned by TRuLer. Our experiments are designed to validate the following claims:

A. TRuLer demonstrates superior scalability compared to the existing transfer rule learner TRL and many standard rule learners.

B. The accuracy of TRuLer for link prediction is competitive compared to existing rule learners, and transferred rules can be used to complement state-of-the-art rule learners to further enhance their link prediction accuracy.

C. TRuLer's performance is robust with varying source rules and on different domains.

## 5.1 Datasets and Baselines

The source rules are obtained from FB15K [3] (for validating Claims A and B), NELL [5] and WN18 [4] (for C), which are all medium-sized KGs that can be easily handled by most existing rule learners. For target KGs, we use three large-scale general-purpose KGs YAGO2s [28], Wikidata [32], and DBpedia 3.8 [1] (for A and B), which are considered challenging for existing rule learners [22]. To verify the effectiveness of our rule transfer approach over KGs of different domains, we also use DRKG [13] and ProteinKG25 [35] as target KGs (for C). DRKG is a comprehensive biological KG that integrates data from six existing databases including DrugBank, Hetionet, GNBR, String, IntAct, and DGIdb. ProteinKG25 is a large-scale KG dataset with aligned descriptions and protein sequences respectively to terms in the GO ontology and protein entities.

The statistics of the datasets are given in Table 1, in which the numbers of entities (#Entity), predicates (#Predicate), training triples (#Train), and test triples (#Test) are recorded.

**Table 1: Statistics of datasets.**

| Dataset | #Entity | #Predicate | #Train | #Test |
|---|---|---|---|---|
| FB15K | 15K | 1345 | 592K | - |
| NELL | 509K | 833 | 760K | |
| WN18 | 41K | 18 | 146K | |
| YAGO2s | 2.2M | 37 | 3.7M | 206K |
| Wikidata | 3.1M | 430 | 7.6M | 420K |
| DBpedia 3.8 | 3.1M | 650 | 9.9M | 552K |
| DRKG | 97K | 107 | 5.3M | 294K |
| ProteinKG25 | 612K | 65 | 8.6M | 481K |

As we want to evaluate the quality of learned rules via link prediction, we divide the target KGs into 90% train, 5% test, and 5% validate sets. Note that a KG here is large and thus a test set or validate set consisting of 5% of the dataset is already large enough. For instance, the link prediction queries that can be generated from the test set of DBpedia are around 552K. Rules are learned on the train set to perform link prediction on the test set.

To our best knowledge, two preliminary approaches TRL [21] and TL-ERMSD [29] to rule transfer learning are reported in the literature. As we were unable to access the latter, TRL is the only transfer rule learner that can be compared. Besides TRL, we also compare our TRuLer with standard rule learners AMIE+ [10], AMIE3 [16], AnyBURL [19], RNNLogic [24], RLogic [9], and RLvLR [22].

The experiments were conducted on a server with Intel Xeon CPU E5-4603 at 2.00GHz (four threads) and with 55GB of RAM, running CentOS 7.9.2009.

## 5.2 Scalability of Rule Transfer

To evaluate the scalability of TRuLer, we transfer rules learned by existing rule learners on FB15K and compare the numbers and coverage of transferred rules with rules directly learned by the same rule learner on the target KG.

RNNLogic [24] and RLogic [9] are not developed to handle large KGs. Therefore, we will compare TRuLer with them in the next section to directly evaluate the performance of the rules generated by these two methods in the link prediction task. As systematic rule searches may take over a week to complete rule learning, we set a 24-hour limit for each rule learner. For TRuLer, the total time includes PCG construction, predicate mapping, rule transfer, and rule validation times. As PCG construction and predicate mapping together only took less than a minute, we recorded both the rule transfer time and the total time in the form of X/Y. For TRL, the rule transfer is intertwined with the rule validation for each predicate, and we could not separate the rule transfer time from the total time for TRL.

The experimental results are shown in Table 2. Following the commonly used thresholds in the literature, we set SC ≥ 0.1 and HC ≥ 0.01. We record the percentages of predicates covered by the learned rules (%P), i.e., the percentages of predicates that occur in the heads of learned rules, and the numbers of rules (#R). All the times are in hours.

We can see that in a reasonable time frame (less than 5 hours in most of the cases), TRuLer can learn a significant number of rules with a good coverage of predicates (above 60% in most of the cases) in the target KGs, whereas other rule learners struggle to learn a similar number of rules with a similar predicate coverage in one day. In particular, TRuLer significantly outperforms transfer rule learner TRL.

While the performance of TRuLer depends on the amount and quality of source rules, its performance is rather robust regarding time efficiency and rule learning capability. For example, the rules learned by TRuLer on Wikidata and DBpedia 3.8 are impacted by the source rules produced by AMIE+ (compared to those by RLvLR and AnyBURL), TRuLer was able to learn a comparable amount of rules within 3 hours as AMIE+ in a day on those KGs. AMIE3 performs well on DBpedia 3.8 and Wikidata, but the number of rules learned per hour is still slightly lower than that of TRuLer.

Also, while the numbers of rules learned by RLvLR on Wikidata and DBpedia 3.8 are comparable to TRuLer, their predicate coverage is significantly lower. This means RLvLR learned more rules for each predicate but also took too long on individual predicates, which makes it infeasible to finish learning for all predicates.

## 5.3 Examples of Learned Rules

Moreover, we examine a few examples of predicate mapping and transferred rules. Table 3 shows the original and transferred rules with their standard confidence, with predicates arranged in the same order to show their correspondence in the mappings.

The rules transferred are all cross-domain and fall into several interesting patterns. Also, the rules are intuitive or at least make some sense. For instance, the rule

$$playsFor(x, y) \rightarrow isAffiliatedTo(x, y)$$

**Table 2: Scalability of TRuLer compared with other (transfer) rule learners.**

| Approach | Source Rules | | YAGO2s | | | Wikidata | | | DBpedia 3.8 | | |
|---|---|---|---|---|---|---|---|---|---|---|---|
| | From | #R | %P | #R | Time (h) | %P | #R | Time (h) | %P | #R | Time (h) |
| AMIE+ | - | - | 32.4 | 35 | 24 | 12.8 | 63 | 24 | 6.9 | 69 | 24 |
| AMIE3 | - | - | 10.8 | 4 | 24 | 51.4 | 1052 | 24 | 56.2 | 1870 | 24 |
| RLvLR | - | - | 70.3 | 176 | 24 | 2.8 | 759 | 24 | 1.1 | 428 | 24 |
| TRL | AMIE+ | 66K | 5.4 | 2 | 24 | 0.5 | 3 | 24 | 0.2 | 1 | 24 |
| TRL | RLvLR | 79K | 13.5 | 9 | 24 | 1.2 | 44 | 24 | 0.2 | 1 | 24 |
| TRL | AnyBURL | 200K | 8.1 | 4 | 24 | 0.9 | 20 | 24 | 0.2 | 1 | 24 |
| TRuLer (Ours) | | | 83.8 | 34 | 1.7/1.8 | 5.1 | 47 | 1.5/3.0 | 3.0 | 29 | 1.0/3.0 |
| TRuLer (Ours) | | | 100 | 250 | 4.7/4.7 | 85.1 | 502 | 0.8/2.3 | 67.5 | 629 | 0.4/2.4 |
| TRuLer (Ours) | | | 100 | 316 | 6.5/6.5 | 68.1 | 700 | 2.6/4.1 | 43.8 | 444 | 1.2/3.2 |

**Table 3: Example rules learned by TRuLer.**

| Source: FB15K | | Target: YAGO2s | |
|---|---|---|---|
| $involved\_in\_merger(x,y) \rightarrow assets\_owned(x,y)$ | 0.14 | $playsFor(x,y) \rightarrow isAffiliatedTo(x,y)$ | 0.99 |
| $basketball\_roster(x,z) \wedge roster(w,z)$ $\wedge sports\_roster(w,y) \rightarrow sports\_roster(x,y)$ | 0.52 | $isInterestedIn(x,z) \wedge isKnownFor(w,z)$ $\wedge isCitizenOf(w,y) \rightarrow isCitizenOf(x,y)$ | 0.33 |
| $phone\_number(x,z) \wedge time\_zones(w,z)$ $\wedge rent50\_3(w,y) \rightarrow retained\_earnings(x,y)$ | 0.41 | $worksAt(x,z) \wedge isLeaderOf(w,z)$ $\wedge livesIn(w,y) \rightarrow isCitizenOf(x,y)$ | 0.23 |
| Source: FB15K | | Target: Wikidata | |
| $pro\_athletes(x,y) \rightarrow sports\_played\_professionally(x,y)$ | 0.99 | $consists\_of(x,y) \rightarrow part\_of(x,y)$ | 0.22 |
| $celebrity\_friends(x,z) \wedge education\_institution(z,y)$ $\rightarrow education\_institution(x,y)$ | 0.05 | $shares\_border\_with(x,z) \wedge country(z,y)$ $\rightarrow country(x,y)$ | 0.95 |
| $gdp\_real(x,z) \wedge gni\_in\_ppp\_dollars(w,z)$ $\wedge athletes(w,y) \rightarrow olympics\_participated\_in(x,y)$ | 0.07 | $spouse(x,z) \wedge noble\_family(w,z)$ $\wedge place\_of\_birth(w,y) \rightarrow place\_of\_death(x,y)$ | 0.17 |
| Source: FB15K | | Target: DBpedia | |
| $award\_nominee(x,y) \rightarrow award\_winner(x,y)$ | 0.35 | $draftTeam(x,y) \rightarrow formerTeam(x,y)$ | 0.37 |
| $awards\_won(x,z) \wedge winners(z,y)$ $\rightarrow award\_nominations(x,y)$ | 0.03 | $league(x,z) \wedge country(z,y) \rightarrow birthPlace(x,y)$ | 0.31 |
| $episode\_performances(x,z) \wedge religious\_practice(z,w)$ $\wedge interests(w,y) \rightarrow draft\_picks(x,y)$ | 0.16 | $primeMinister(x,z) \wedge deathPlace(z,w)$ $\wedge leaderParty(w,y) \rightarrow party(x,y)$ | 0.20 |

says if a player $x$ plays for a team $y$ then $x$ is affiliated to $y$. And the rule

$$shares\_border\_with(x,z) \wedge country(z,y) \rightarrow country(x,y)$$

says if two regions $x$ and $z$ share a border and $z$ is part of a country $y$ then $x$ is likely to be part of $y$.

## 5.4 Quality of Transferred Rules in Reasoning

The second set of experiments aim to evaluate the quality of learned rules via link prediction. The evaluation of link prediction is to show the quality of learned rules, so TRuLer is not compared with embedding-based link prediction models.

In this set of experiments, we show that the rules learned by TRuLer can achieve a good level of accuracy in link prediction, and the learned rules can complement state-of-the-art rule learners to enhance their performance in link prediction. For TRuLer, we used the transferred rules from source rules generated by AnyBURL, due to its outstanding link prediction accuracy reported in the literature. As for evaluation metrics, we adopt Mean Reciprocal Rank (MRR) and Hits@10 (H@10), which are the most widely used metrics for link prediction. MRR is the average of the reciprocal ranks of the desired entities and Hits@10 is the percentage of desired entities being ranked among the top ten.

We compare the rules learned by TRuLer with those learned by AMIE3, RLvLR, RLogic, RNNLogic, and AnyBURL, respectively. Besides, we also combine the rules learned by TRuLer with those from existing rule learners. As the rules learned by AMIE3 and RLvLR have an extremely low predicate coverage on Wikidata and DBpedia 3.8, we used link prediction queries generated with predicates covered by all the six rule learners. That is, we compare with rule learners on those predicates they all have learned rules to cover: 20 predicates for YAGO2s, 18 for Wikidata, and 7 for DBpedia 3.8. Note that while it only involves a small number of predicates, the total number of queries for each KG is still high: 75K for YAGO2s, 44K for Wikidata, and 11K for DBpedia 3.8. Results are summarized in Table 4.

This experiment shows that TRuLer performs well in link prediction, outperforming AMIE3, RLogic, RNNLogic, and RLvLR in most of the cases, which shows the transferred rules are of high quality. This shows TRuLer can effectively explore useful rule patterns for transfer and learn rules for individual predicates with higher quality compared to exhaustive rule learners. Moreover, the rules learned by TRuLer can complement those directly learned by a rule learner to improve their performance in link prediction. This shows the transferred rules capture useful rule patterns that are missed by state-of-the-art rule learners.

**Table 4: Comparison on link prediction.**

| Models | YAGO2s | | Wikidata | | DBpedia 3.8 | |
|---|---|---|---|---|---|---|
| | MRR | H@10 | MRR | H@10 | MRR | H@10 |
| RLogic | 0.009 | 0.011 | 0.023 | 0.029 | 0.036 | 0.042 |
| RNNLogic | 0.044 | 0.051 | 0.012 | 0.027 | 0.020 | 0.033 |
| AMIE3 | 0.048 | 0.048 | 0.014 | 0.016 | 0.040 | 0.055 |
| RLvLR | 0.136 | 0.149 | 0.038 | 0.041 | 0.027 | 0.036 |
| AnyBURL | 0.138 | 0.167 | 0.070 | 0.093 | **0.051** | 0.085 |
| TRuLer | 0.180 | 0.197 | 0.041 | 0.050 | 0.043 | 0.059 |
| TRuLer + AMIE3 | 0.181 | 0.198 | 0.061 | 0.065 | **0.051** | 0.068 |
| TRuLer + RLVLR | **0.194** | 0.211 | 0.069 | 0.077 | 0.044 | 0.060 |
| TRuLer + AnyBURL | 0.190 | **0.212** | **0.080** | **0.097** | **0.051** | **0.091** |

## 5.5 Further Analysis

We conduct another three sets of experiments to analyse the impact of mapping methods, source rules, and domain knowledge on rule transfer.

*5.5.1 Mapping Methods.* To analyse the effectiveness of our predicate mapping approach, we replace the predicate mapping module in TRuLer with ontology alignment methods. We use existing ontology alignment methods SEU [18] and FGWEA [31] to replace the predicate mapping module in TRuLer, and the comparison results are shown in Table 5.

**Table 5: Comparison with ontology alignment algorithms.**

| Mapping Methods | YAGO2s | | Wikidata | | DBpedia 3.8 | |
|---|---|---|---|---|---|---|
| | %p | #R | %p | #R | %p | #R |
| SEU | 24.3 | 10 | 47.7 | 260 | 51.7 | 532 |
| FGWEA | 56.8 | 28 | 23.3 | 107 | 41.1 | 346 |
| Ours | **100** | **265** | **86.5** | **671** | **78.9** | **1024** |

It can be seen that our mapping method significantly outperforms ontology alignment methods. This shows our mapping method preserving graph patterns is more suitable for rule transfer than alignment methods based on lexical or semantic similarities.

*5.5.2 Source Rules.* To analyse the impact of source rules on rule transfer, we use source rules obtained from various KGs, namely subsets of FB15K of varying sizes, NELL, and WN18. We obtained subsets of FB15K with increasing sizes, via sampling of 20, 100, 200, 500, and 1000 predicates. Then, we employ RLvLR to learn the source rules. The experimental results are shown in Table 6.

Overall, as the number of predicates increases, the number of rules transferred to the target KG also increases. On the other hand, while the numbers of entities and facts also contribute to the diversity of source rules, they have less impact on the rule transfer, which can be seen by comparing FB15K-20 and WN18, and FB15K-1000 and NELL.

*5.5.3 Domain Knowledge.* To analyse the performance of TRuLer on cross-domain rule transfer, we transfer rules to DRKG and ProteinKG25 that are quite different domain knowledge from the source KG FB15K. The results are shown in Table 7.

**Table 6: Rule transfer with varying source rules.**

| Source KG | YAGO2s | | Wikidata | | DBpedia 3.8 | |
|---|---|---|---|---|---|---|
| | %P | #R | %P | #R | %P | #R |
| WN18 | 59.5 | 53 | 10.2 | 54 | 11.2 | 201 |
| FB15K-20 | 48.6 | 32 | 2.6 | 11 | 2.3 | 18 |
| FB15K-100 | 89.2 | 88 | 14.0 | 72 | 11.2 | 100 |
| FB15K-200 | 97.3 | 141 | 26.5 | 137 | 19.2 | 207 |
| FB15K-500 | 100 | 224 | 59.3 | 394 | 50.6 | 552 |
| NELL | 100 | 255 | 84.7 | 539 | 67.7 | 987 |
| FB15K-1000 | 100 | 261 | 84.2 | 645 | 73.5 | 981 |
| FB15K | **100** | **265** | **86.5** | **671** | **78.9** | **1024** |

**Table 7: Cross-domain rule transfer.**

| Approach | DRKG | | | ProteinKG25 | | |
|---|---|---|---|---|---|---|
| | %P | #R | Time (h) | %P | #R | Time (h) |
| AMIE3 | 75.7 | 1872 | 24 | 86.2 | 291 | 24 |
| RLvLR | 9.3 | 313 | 24 | 9.23 | 56 | 24 |
| TRL | 12.0 | 117 | 24 | 11.7 | 47 | 24 |
| TRuLer | 97.2 | 7997 | 13.33 | 98.5 | 799 | 20 |

We can see that TRuLer performs well in terms of rule quantity and predicate coverage compared to direct rule learning. This demonstrates the effectiveness of our rule pattern transfer approach, which does not rely on a mapping of the semantics between the source and target domains. With the richer structural information in DRKG and ProteinKG25 compared to the previous three KGs, more rules are learned by TRuLer with often a better predicate coverage. On the other hand, rule learners like RLvLR that exhaustively search the rule space struggle on ProteinKG25.

## 6 CONCLUSION

Learning first-order rules on a large-scale KG is a challenging problem. Existing rule learning methods use exhaustive search to construct rules directly from the KG, while transfer rule learning has been rarely explored. In this paper, we have proposed a scalable and effective framework TRuLer for rule learning based on the paradigm of transfer learning. A key component of TRuLer is a novel predicate embedding method that can capture useful structural features for rule transfer. To achieve this goal, we have introduced a predicate alignment mechanism for mapping the predicates between the source and the target KGs. Experimental evaluation shows that TRuLer can handle large-scale KGs such as Wikidata and DBpedia, for rule learning in terms of both quantity and quality (within a reasonable time frame). Also, the combination of TRuLer with a standard rule learner can improve the performance of the standard rule learner within a large-scale KG.

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
