# OpenReview forum: "Transfer Rule Learning over Large Knowledge Graphs"
_ACM.org/TheWebConf/2025/Conference — WWW 2025 Poster_

### Official Review · Reviewer_XykB · 2024-11-29

**Novelty:** 6
**Technical Quality:** 3

**Review:**

This article presents the TRuLer approach for transfer rule learning over knowledge graphs (KG). Given a set of rules mined from a source KG, TRuLer can exploit the structural connections of such rules and transfer it into a target KG resulting in a set of rules defined for the target domain. Those connections are encoded via graphlet degree vectors derived from a set of predicate-correlation graphs, and enriched with a message-passing-like procedure. Those vectors are then mapped in a subsequent process based on the REGAL alignment method. The experimental evaluation reveals that such an approach works on large KGs and can obtain high quality rules with competitive performance on different link prediction benchmarks. It also outperforms existing rule transfer approaches.

1. The proposed method is interesting because it does not make many assumptions about the involved KGs, and capitalizes on the results of a computationally-intensive task run hopefully only once. I think, however, that the article lacks insights on how to exploit TRuler at best. For example, the experimental evaluation does not explain why FB15K was used as source KG. Is it because it has the most predicates? Moreover, how long did it take to mine those initials rules on FB15K? Table 2 is a bit hard to read. What are the meanings of the three "TRuler (Ours)" lines? (Is it TRuler + the methods on the block above?). I think a study of the impact of the source domain (e.g., noisy predicates) and/or the mining extraction settings is mandatory.

2. It is a bit perplexing to see that the confidences of the exemplified rules are that low. While that is consistent with the low scores reported for MRR and H@10, I wonder why the authors did not consider other datasets where rule mining performs a bit better. AnyBURL, for example, reports way higher scores: https://web.informatik.uni-mannheim.de/AnyBURL/#results for standard public benchmark datasets.
It would be more useful for future users of TRuLer to see that the approach can transfer good rules in the source KG into similarly good rules in the target KG. Unfortunately that is neither shown in the experimental section nor discussed in the conclusion.

3. A minor thing. I would recommend the users to clarify the definition of support for \epsilon-neighbour relations. Some of those relations involve three variable positions. Could you please exemplify how to compute the support of an edge TH and an edge LP?

**Questions:**

- What is the impact of the domain KG and the mining extraction settings on the quality of the transferred rules?
- Why did the authors not consider other datasets where rule mining performs a bit better?
- How long did it take to mine the source rules on FB15K? Why only FB15K?
- Could you please exemplify how to compute the support of an edge TH and an edge LP?

**Reviewer Confidence:**

4: The reviewer is certain that the evaluation is correct and very familiar with the relevant literature

**Scope:**

4: The work is relevant to the Web and to the track, and is of broad interest to the community

---

### Official Review · Reviewer_ifey · 2024-12-01

**Novelty:** 4
**Technical Quality:** 4

**Review:**

This paper proposes a novel framework, TRuLer (Transfer Rule Learner), for rule learning in large-scale knowledge graphs (KGs) using transfer learning. The authors address the challenge of transferring rules from a source KG to a target KG by leveraging graph-structural similarities between predicates. The proposed framework introduces a predicate-correlation graph (PCG) and a predicate embedding method based on graphlet degree vectors (GDVs), enabling scalable and effective rule transfer. Extensive experiments demonstrate TRuLer’s scalability, rule quality, and robustness across domains.






Pros
1. Introduces a novel approach combining structural similarity and transfer learning for rule extraction.
2. Efficiently handles large KGs like Wikidata and DBpedia with millions of triples.
3. Performs well across diverse domains, including biological KGs like DRKG and ProteinKG25.


Cons
1. The framework heavily relies on structural similarity for predicate mapping, ignoring semantic relationships between predicates. This could limit its applicability in scenarios where structural similarity alone is insufficient.

2. The rule validation process relies on metrics such as support and confidence, but their effectiveness in filtering noisy rules in sparse or incomplete target KGs is not thoroughly discussed.

3. While the experiments demonstrate scalability on large KGs, the paper does not provide a detailed complexity analysis of the PCG construction, predicate embedding, and rule validation steps.

**Questions:**

1. How does TRuLer handle cases where predicates in the source and target KGs are semantically similar but structurally dissimilar? Could incorporating semantic similarity (e.g., using word embeddings) improve predicate mapping?
2. How does TRuLer ensure the quality of transferred rules in sparse or incomplete target KGs? Have you considered more sophisticated validation techniques?
3. To what extent does the performance of TRuLer depend on the quality and diversity of source rules? How would it perform with noisy or incomplete source rules?
4. Can you provide a detailed complexity analysis of each step in the TRuLer framework (e.g., PCG construction, predicate embedding, rule validation)?
5. Were the source rules used in TRuLer generated under the same conditions as the rules directly learned by baseline methods? If not, how might this affect the comparisons?
6. How would TRuLer handle cases where the transferred rules are not semantically or practically meaningful in the target domain?

**Reviewer Confidence:**

3: The reviewer is confident but not certain that the evaluation is correct

**Scope:**

4: The work is relevant to the Web and to the track, and is of broad interest to the community

---

### Official Review · Reviewer_PBFM · 2024-12-02

**Novelty:** 4
**Technical Quality:** 5

**Review:**

### **Pros**:
1. **Performance**:  The proposed method demonstrates better performance and faster rule learning compared to both transfer and standard rule learners.
2. **Diverse Datasets**: The proposed method has been tested on multiple real-world datasets, ranging from general-purpose datasets to domain-specific ones.

### **Cons**:
1. **Novelty**: The framework feels more like a combination of existing methods with slight tweaks, lacking substantial novelty.
2. **Ablation Study**: The experiment doesn’t include ablation studies, making it difficult to understand the individual impact of each module.
3. **Baseline**: The baseline rule learning models used for comparison seem outdated, and it is missing benchmarks against more recent approaches like "Hi-KnowE: Xie S, Liu R, Wang X, et al. Hierarchical Knowledge-Enhancement Framework for multi-hop knowledge graph reasoning[J]. Neurocomputing, 2024, 588: 127673."

**Questions:**

1. **Ablation Study**:  Can ablation studies be added to clarify the individual contributions of SC, HC metrics, and other modules to the overall performance?
2. **Baseline**:  Why were more recent rule learning models not included in the baseline comparisons? The latest baseline seems to be from 2022.
3. **Method**:  Can the SC and HC-based filtering strategy be widely applied to other knowledge graphs? If rule distribution characteristics change, would the filtering strategy need adjustment?
4. **Figure**:  Figure 3 is not intuitive enough. The small diagrams on the left and right sides of the illustration are difficult to understand and lack detailed explanations, making them hard to interpret, as is the case with Figure 2.

**Reviewer Confidence:**

3: The reviewer is confident but not certain that the evaluation is correct

**Scope:**

4: The work is relevant to the Web and to the track, and is of broad interest to the community

---

### Official Review · Reviewer_qn5f · 2024-12-02

**Novelty:** 6
**Technical Quality:** 7

**Review:**

The work proposes a novel transfer learning framework called TRuLer (Transfer Rule Learner) that addresses the challenge of learning logical rules in large knowledge graphs by leveraging existing rules from smaller source knowledge graphs. Rather than learning rules from scratch through computationally expensive searches, the framework introduces a predicate mapping method based on graph-structural similarity to effectively transfer established rules between knowledge graphs. The key technical contribution is a predicate-correlation graph representation that captures how predicates are structurally related, allowing rules to be adapted from Source to Target graphs while preserving relevant patterns.

Regarding quality:
To the best of my knowledge the paper seems well written, it clearly presents the aim, it proceeds in order, it provides reasons for the approach choices (which may or may not be shared, but still a reason is always provided), and it supports the approach with sufficient formalisms.

Regarding clarity:
The "Approach" part is pretty dense, but such is the nature of 9 paper pages, the work keeps the approach understandable while including technical formalisms.

Regarding originality: to the best of my knowledge the work tackles an interesting matter, and it does so applying more common transfer learning techniques on a structural level, testing an original approach to an open problem.

Regarding significance: the paper is significant and could constitute scientific advancements.

Pros:
- sound, scalable, and conceptually interesting approach
- good results, replicable experiments
- good work presentation

Cons:
- the usage of 90% of the triples for training and only 5% for testing could lead to some potential problems, as detailed in the questions section

As a final remark: I think that what you are actually doing is known in classical ontology modeling as "reification". It means in this case that you consider each edge as a node, in order to be able to predicate something about it. Nevertheless, the fact that you don't call it like that, does not diminish the quality of your work, and instead it reinforces your approch. Mentioning this matter could ground your work in some sound practices of knowledge graphs modeling design.

**Questions:**

The usage of 90% of the kg as training set is not per se wrong, given that we are not dealing with classical machine learning tasks, but still in my opinion there are some blindspots which could be addressed:

1. Using only 5% of the kg as test case, are all the rules learnt from the 90% instantiated at least one time in the test set? If so, that's ok, it confirms what you say that we are still dealing with thousands of triples. If not: careful that this points in the direction of not being really able to test all rules, but only the most common ones.

2. Regarding this same issue: how do you deal with long-tail issues, which in knowledge graphs could be relevant?

3. Finally, on a side note, this unusual split could make it difficult to compare this paper with future works and benchmarks.
It could be interesting to keep your evaluation as it is, but to include as additional materials, some experiments on a more classic 80/20 or even 70/30 and see if, as expected, the performances decrease, and how much.

**Reviewer Confidence:**

3: The reviewer is confident but not certain that the evaluation is correct

**Scope:**

4: The work is relevant to the Web and to the track, and is of broad interest to the community